# Structural basis for the type I-F Cas8-HNH system

Xuzichao Li [1,2,6], Yanan Liu [3,6], Jie Han[1,4,6], Lingling Zhang[1,6], Zhikun Liu [1], Lin Wang[1], Shuqin Zhang[1], Qian Zhang[1], Pengyu Fu[5], Hang Yin[5], Hongtao Zhu [3✉] & Heng Zhang [1,2✉]

## Abstract

The Cas3 nuclease is utilized by canonical type I CRISPR-Cas systems for processive target DNA degradation, while a newly identified type I-F CRISPR variant employs an HNH nuclease domain from the natural fusion Cas8-HNH protein for precise target cleavage both in vitro and in human cells. Here, we report multiple cryo-electron microscopy structures of the type I-F Cas8-HNH system at different functional states. The Cas8-HNH Cascade complex adopts an overall G-shaped architecture, with the HNH domain occupying the C-terminal helical bundle domain (HB) of the Cas8 protein in canonical type I systems. The Linker region connecting Cas8-NTD and HNH domains adopts a rigid conformation and interacts with the Cas7.6 subunit, enabling the HNH domain to be in a functional position. The full R-loop formation displaces the HNH domain away from the Cas6 subunit, thus activating the target DNA cleavage. Importantly, our results demonstrate that precise target cleavage is dictated by a C-terminal helix of the HNH domain. Together, our work not only delineates the structural basis for target recognition and activation of the type I-F Cas8-HNH system, but also guides further developments leveraging this system for precise DNA editing.

**Keywords** CRISPR-Cas; Type I-F System; Cas8-HNH; HNH Domain; Genome Editing
**Subject Categories** DNA Replication, Recombination & Repair; RNA Biology; Structural Biology

## Introduction

CRISPR-Cas systems provide an RNA-guided adaptive immune response for prokaryotes to defend against invading mobile gene elements (Barrangou and Marraffini, 2014; Sorek et al, 2013; Wiedenheft et al, 2012). They are classified into two major classes and six subtypes (Makarova et al, 2020; Shmakov et al, 2017). Class 1 CRISPR systems utilize multi-subunit effector complex for degradation of nucleic acid targets and can be further divided into three subtypes (type I, III, and IV). Class 2 CRISPR systems utilize a single Cas effector to destroy nucleic acid targets, and are further divided into type II, V, and VI subtypes (Abudayyeh et al, 2017; Gasiunas et al, 2012; Staals et al, 2014; Staals et al, 2013; Zetsche et al, 2015). Class 1 type I systems are the most widely distributed in bacteria and archaea, utilizing a crRNA-guided Cascade (CRISPR-associated complex for antiviral defense) surveillance complex and a signature helicase-nuclease Cas3 protein for sequential target recognition and degradation process (Csorgo et al, 2020; Shangguan et al, 2022; Xiao et al, 2018; Yoshimi et al, 2022). Once the type I surveillance complex identifies the invading DNA through the protospacer adjacent motif (PAM) sequence, the double strands of target DNA are unwound and engaged in the Cascade backbone, forming an R-loop bubble (Guo et al, 2017; Hayes et al, 2016; Rutkauskas et al, 2015; Szczelkun et al, 2014; van Erp et al, 2015). Structural snapshots of type I-E and I-F systems show that the Cas3 helicase-nuclease is only captured by the Cascade complex containing a full R-loop (Guo et al, 2017; Hayes et al, 2016; Rollins et al, 2019; Xiao et al, 2018). Cas3 initially nicks the R-loop, switches along the non-target strand from the initial cleavage site, and subsequently engages in processive DNA degradation (Mulepati and Bailey, 2013; Redding et al, 2015; Rollins et al, 2017; Xiao et al, 2017). Type I systems have been commonly used for gene editing and have been demonstrated to alter gene expression and produce long-range deletions in human cells (Cameron et al, 2019; Chen et al, 2020; Osakabe et al, 2021; Zheng et al, 2019).

Recently, a newly identified type I-F variant, known as the Cas8-HNH system, was discovered to lack the Cas3 helicase-nuclease gene in its CRISPR loci (Altae-Tran et al, 2023). Instead, an HNH nuclease domain is naturally fused with one of the Cascade backbone components, Cas8, generating double-strand breaks in target DNA molecules. In contrast to the previously characterized type I systems known for uncontrolled cleavage and preferential targeting of dsDNA, this type I-F variant has been demonstrated to precisely cleave both double-stranded DNA (dsDNA) and single-stranded DNA (ssDNA) targets, highlighting its potential for innovative gene editing applications.

To elucidate the molecular mechanisms of target DNA cleavage by the type I-F Cas8-HNH system, we reconstituted the Cascade

[1]Tianjin Institute of Immunology, State Key Laboratory of Experimental Hematology, International Joint Laboratory of Ocular Diseases (Ministry of Education), Key Laboratory of Immune Microenvironment and Disease (Ministry of Education), The Province and Ministry Co-sponsored Collaborative Innovation Center for Medical Epigenetics, School of Basic Medical Sciences, Tianjin Medical University, Tianjin 300070, China. [2]Department of Biochemistry and Molecular Biology, Tianjin Key Laboratory of Cellular Homeostasis and Disease, School of Basic Medical Sciences, Tianjin Medical University, Tianjin, China. [3]Beijing National Laboratory for Condensed Matter Physics, Institute of Physics, Chinese Academy of Sciences, Beijing, China. [4]Department of Anatomy, School of Basic Medical Sciences, Tianjin Medical University, Tianjin, China. [5]Department of Pharmacology, School of Basic Medical Sciences, Tianjin Medical University, Tianjin 300070, China. [6]These authors contributed equally: Xuzichao Li, Yanan Liu, Jie Han, Lingling Zhang. ✉E-mail: hongtao.zhu@iphy.ac.cn; zhangheng134@gmail.com

complex and obtained multiple cryo-EM structures representing the target-free, partial R-loop, full R-loop and ssDNA-bound states. The Cas8-HNH Cascade complex adopts an overall G-shaped architecture, with the fused HNH nuclease domain anchored at the center of the complex by a linker region. Structural and biological analyses reveal that full R-loop formation is crucial for HNH nuclease activation. The C-terminal helix of the HNH domain is utilized to detect the target DNA-crRNA hybrid and facilitate the conformational changes of the HNH domain required for the nuclease activation. Mismatches in the protospacer sequence of the target DNA are not tolerated, particularly at the PAM-distal positions. These features efficiently avoid the mistargeting of the type I-F Cas8-HNH system, indicating a highly accurate gene editing tool. These structural and biological results will guide further developments of the type I-F Cas8-HNH system, leveraging this new DNA targeting system for precise gene-editing applications.

## Results

### Biochemical characterization of the type I-F Cas8-HNH system

The newly identified type I-F variant, termed the Cas8-HNH system, is reported to exhibit precise RNA-guided target DNA cleavage activity (Altae-Tran et al, 2023). Despite lacking a *cas3* gene in this system, the Cas8 subunit is fused to another nuclease domain, HNH, via a Linker region (residues 217–235), referred to as Cas8-HNH protein (Fig. 1A,B). We purified the type I-F Cas8-HNH system complex consisting of Cas5, Cas6, Cas7, and Cas8-HNH subunits and a crRNA from *Selenomonas* sp. isolate RGIG9219 and carried out in vitro dsDNA and ssDNA cleavage assays. The Cas8-HNH Cascade complex was found to efficiently degrade both the dsDNA and ssDNA targets (Appendix Fig. S1A–C), while the catalytic mutant H305A in the HNH nuclease domain abolished the target DNA cleavage (Appendix Fig. S1A,B). Given that the HNH domain cuts DNA in a metal ion-dependent manner (Joshi et al, 2015; Zhang et al, 2017), we conducted the target DNA cleavage assay in the presence of various metal ions (Appendix Fig. S1D). The divalent ions $Mg^{2+}$ or $Mn^{2+}$ could enable the efficient target cleavage, whereas $Zn^{2+}$, $Fe^{3+}$, or $Ca^{2+}$ failed to activate the nuclease activity of type I-F Cas8-HNH system, indicating a $Mg^{2+}$ or $Mn^{2+}$ dependent nuclease activity, consistent with the previously reported HNH domains (Joshi et al, 2015; Zhang et al, 2017).

### Cryo-EM structure of the type I-F Cas8-HNH system in a target-free state

To elucidate the molecular basis of the type I-F Cas8-HNH system, we reconstituted the Cas8-HNH Cascade complex with a 68-nucleotide (nt) crRNA. Utilizing single-particle cryo-electron microscopy, we solved the structure of the Cascade-crRNA ribonucleoprotein (RNP) complex at a resolution of 3.6 Å (Fig. 1C,D; Appendix Fig. S2). The RNP complex displays an overall G-shaped architecture, comprising single copies of Cas5, Cas6, and Cas8-HNH subunits, six copies of Cas7 molecules, and a crRNA molecule. Almost all the

residues of these subunits were well-resolved except for several loop regions, possibly due to the intrinsic flexibility (Appendix Fig. S3). The EM density allowed us to trace the majority of crRNA containing an 8-nt 5′-handle sequence, a 32-nt spacer-derived region (termed as spacer), and a 28-nt 3′-stem loop (Fig. 1E; Appendix Fig. S3G). Six Cas7 subunits interlock with each other in a head-to-tail manner and wrap the crRNA, forming the main backbone of the G-shaped complex, reminiscent of the typical type I-F system (Appendix Fig. S4). The 3′-stem loop of crRNA is capped by Cas6 protein at the head of the RNP complex, while the Cas5 subunit, along with the N-terminal domain (NTD) of Cas8-HNH protein, recognizes the 5′-handle of the crRNA, masking the tail of the backbone. The C-terminal HNH domain of Cas8-HNH protein, which occupies the HB domain of Cas8 protein in the canonical type I-F system, is located at the center of the G-shaped surveillance complex. Analysis of the protein surface potential showed a small positively charged binding groove on the HNH domain, which may be responsible for the target DNA binding (Appendix Fig. S5A). To test this hypothesis, we next selected a series of charged and aromatic residues lining this positively charged groove for mutagenesis and functional studies (Fig. 1F). As expected, alanine replacement of these basic or aromatic residues, such as Y271A/R274A/H275A/R277A, almost abolished the target DNA cleavage by the type I-F Cas8-HNH system, indicating the importance of these residues in HNH nuclease activity (Fig. 1G; Appendix Fig. S5B).

### Cryo-EM structure of type I-F Cas8-HNH system in target-bound states

To further reveal the DNA recognition and cleavage mechanisms by type I-F Cas8-HNH system, we reconstructed the Cascade-crRNA-target DNA complex with a 56-base-pair (bp) dsDNA target containing a GCG PAM and determined two cryo-EM structures representing the full R-loop state and the partial R-loop state at resolutions of 2.51 and 2.57 Å, respectively (Fig. 2A–C; Appendix Fig. S6). In the full R-loop state, the whole 32-nt protospacer sequence on the target strand (TS) was observed to pass through a positively charged channel of the interlocked Cas7 subunits (Appendix Fig. S6H). In the protospacer sequence, every five consecutive nucleotides hybridize with the crRNA, followed by a single nucleotide kinked by a β-hairpin protruding from the Cas7 subunits, forming a periodic "5 + 1" pattern. Five periodic "5 + 1" segments are followed by another separate 2-nt sequence base-paired to the crRNA (Fig. 2B). In the partial R-loop state, only the TS sequence of the first four periodic segments and three base-paired nucleotides in the fifth periodic segment could be well traced according to the EM density (Fig. 2A,B), representing an intermediate state in the formation of the R-loop bubble. Furthermore, we also determined the cryo-EM structure of the Cas8-HNH Cascade-crRNA-ssDNA complex at a resolution of 2.9 Å with a 56-nt targeting ssDNA (Fig. 2D; Appendix Fig. S7). The whole 32-nt protospacer sequence of ssDNA was clearly traced. The complex in the ssDNA-bound state closely resembles that in the full R-loop state, with the exception of the hook region of the Cas8, which undergoes an outward rotation upon PAM binding (Appendix Fig. S8). We therefore focused on the structural analysis of the target-bound complex in the full R-loop state due to its high resolution.

                                                                 

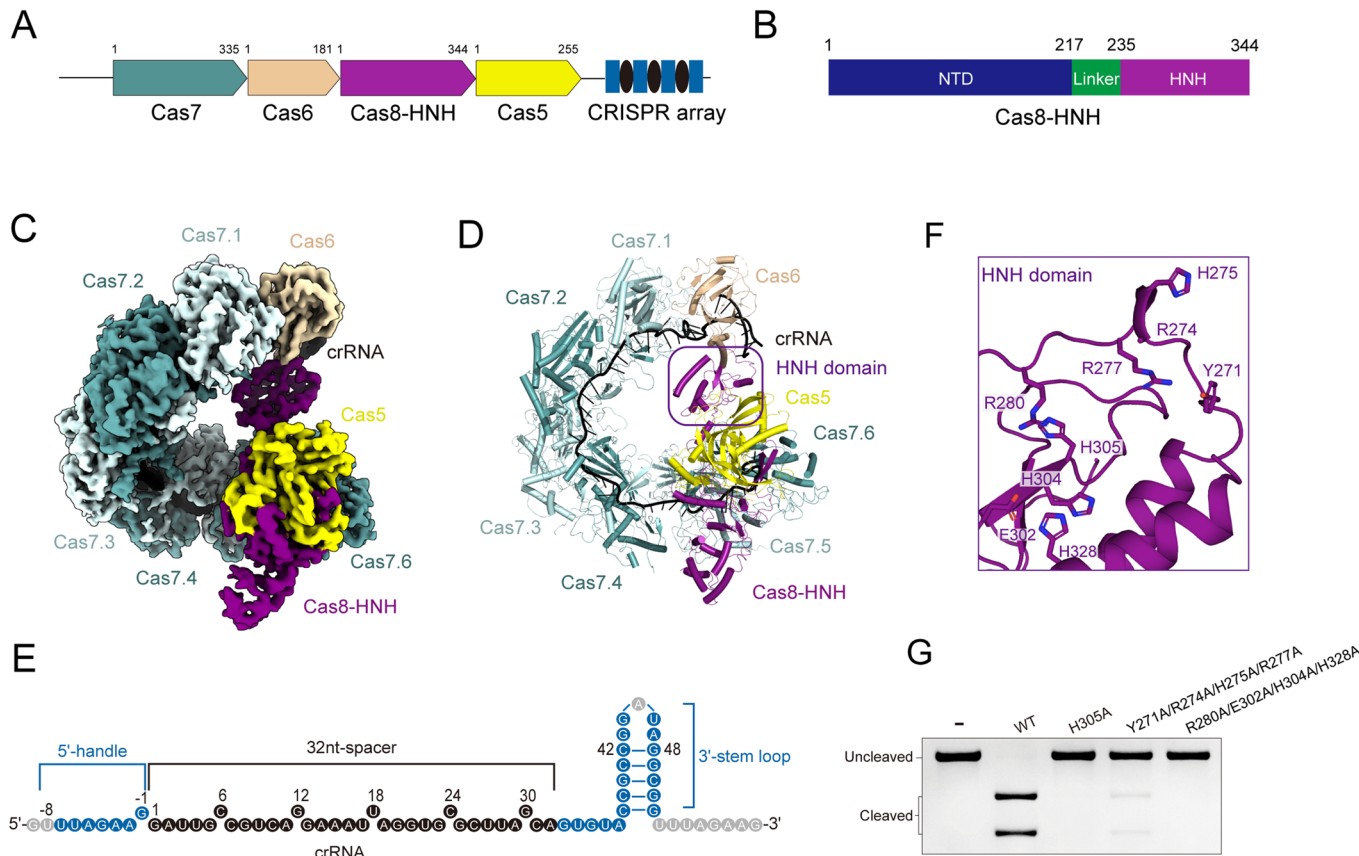

**Figure 1. Cryo-EM structure of type I-F Cas8-HNH system in target-free state.**

(A) Gene architectures of type I-F Cas8-HNH system. (B) Domain organization of Cas8-HNH protein. The N-terminal domain (NTD), Linker, and HNH domain are colored dark blue, green, and purple, respectively. (C) Cryo-EM density map of the Cas8-HNH ribonucleoprotein (RNP) complex. Cas6, Cas8, Cas5 subunits, and crRNA are shown in wheat, purple, yellow, and black, respectively. Six copies of Cas7 are colored with different shades of cyan. (D) Atomic model of type I-F Cas8-HNH RNP complex. The same color scheme as in 1C. The HNH domain is marked with a purple box. (E) Schematic representation of crRNA. Repeat-derived 5'-handle sequence and 3'-stem loop of crRNA are represented by blue circles. The 32-nucleotide (nt) spacer sequence is represented by black circles. Unbuilt nucleotides in the cryo-EM structure are represented by gray circles. (F) Close-up view of the positively charged DNA binding pocket on the HNH domain. A series of charged and aromatic residues lining the binding pocket are shown as sticks. The catalytic residue His305 is also shown in stick representation. (G) In vitro dsDNA cleavage assay of type I-F Cas8-HNH Cascade variants bearing mutations in HNH nuclease domain. Key residues responsible for target DNA binding are mutated. The gel represents three independent replicate experiments. Source data are available online for this figure.

## PAM recognition and DNA engagement

Type I interference complexes distinguish self from non-self targets through PAM recognition. In the target-bound state (full R-loop state, unless indicated otherwise), a 5-bp PAM-containing duplex (PAM duplex) was unambiguously traced based on the high-quality EM density. A lysine finger (Lys85) on the NTD stacks against the PAM duplex and forms a hydrogen bond with the base of -3dG$^{NTS}$ (Appendix Fig. S9A). The adjacent acidic residue (Asp83) forms a hydrogen bond with the -2'dG base moiety, possibly specifying a guanine at the -2′ site of the TS. Furthermore, two basic residues, Lys42 and Arg88, interact with the phosphate backbone of -3′dC$^{TS}$ and -1dG$^{NTS}$ through electrostatic contacts, respectively. Notably, mutations of these PAM-interacting residues, such as K85A/R88A, severely compromised target DNA cleavage, indicating the importance of PAM recognition in target cleavage (Appendix Fig. S9B).

Compared to the target-free state, a hook region on the NTD of Cas8-HNH protein rotates inward to stabilize the PAM duplex in the dsDNA-bound state (Appendix Fig. S9C). An Asn-wedge helix (residues 193-198) protrudes into the target DNA and unwinds the double strands after the PAM site (Appendix Fig. S9C). Particularly, two nucleotides 1dG$^{NTS}$ and 2dA$^{NTS}$, after the PAM sequence were clearly traced in the cryo-EM density. Two polar residues, Asn193 and Ser196, on the Asn-wedge are hydrogen bonded with the base of 1dG$^{NTS}$, and the base of 2dA$^{NTS}$ is coordinated by Ser204 (Appendix Fig. S9A). Mutation of these polar residues prevented the Cas8-HNH-mediated target DNA cleavage, indicating the importance of the purine recognition at positions 1 and 2 on NTS for HNH nuclease activity (Appendix Fig. S9B). It has been reported that the type I-F Cascade complex from *Pseudomonas aeruginosa* recognizes the CC PAM sequence and unwinds DNA through a Lys-wedge loop (Guo et al, 2017; Rollins et al, 2019). In contrast, the GCG PAM is mainly recognized by a Lysine finger, and the DNA unzipping is mainly mediated by an Asn-wedge helix in the Cas8-HNH system, indicative of a distinct recognition and engagement mode of target dsDNA.

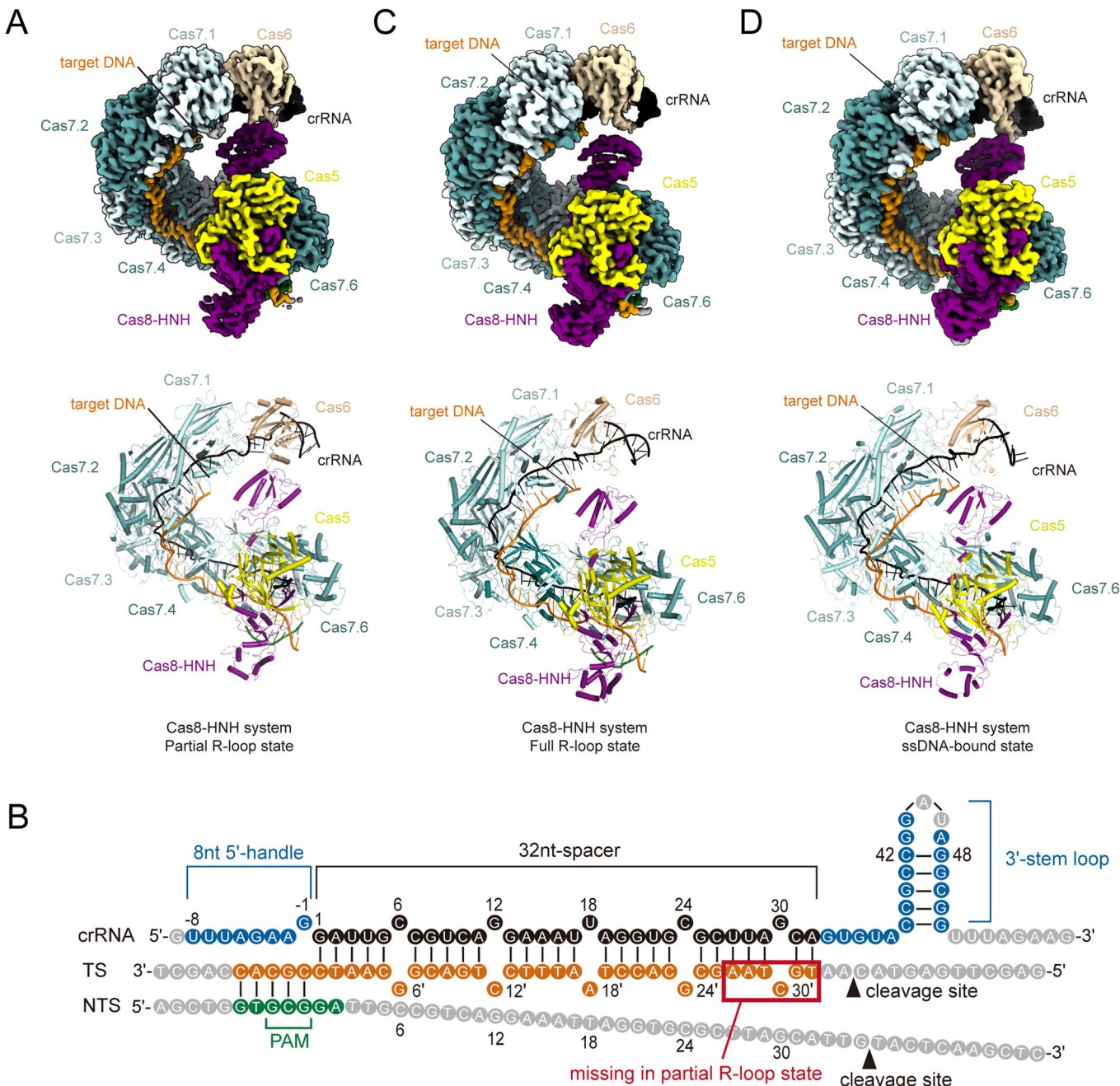

**Figure 2. Cryo-EM structure of type I-F Cas8-HNH system in target-bound states.**

(A) Cryo-EM density map (upper panel) and atomic model (lower panel) of type I-F Cas8-HNH system in partial R-loop states. The TS and NTS are colored orange and forest green, respectively. (B) Schematic diagram of the crRNA and target DNA. The traced nucleotides from TS and NTS in the structure are represented by orange and green circles, respectively. The target DNA is fully complementary to the crRNA spacer sequence. The 27′–32′ nucleotides on the TS are missing in the partial R-loop state and are marked with a red box. (C, D) Cryo-EM density map (upper panel) and atomic model (lower panel) of type I-F Cas8-HNH system in full R-loop and ssDNA-bound states

## The Linker region is responsible for the functional localization of the HNH domain

The HNH nuclease domain is tethered to the Cascade backbone by the Linker region of the Cas8-HNH protein (Fig. 1B–D). In the present cryo-EM structures, the Linker part folds into a short helix and

interacts primarily with the Cas7.6 subunit (Fig. 3A), suggesting an anchor role of the Linker region in HNH location. Specifically, Lys222 of the Linker forms a hydrogen bond with Glu244 on the Cas7 subunit. In addition, two leucine residues (Leu224 and Leu228) from the Linker, contact with Leu48 and Pro46 of the Cas7.6 subunit through hydrophobic interactions (Fig. 3A). Interestingly, the interactions

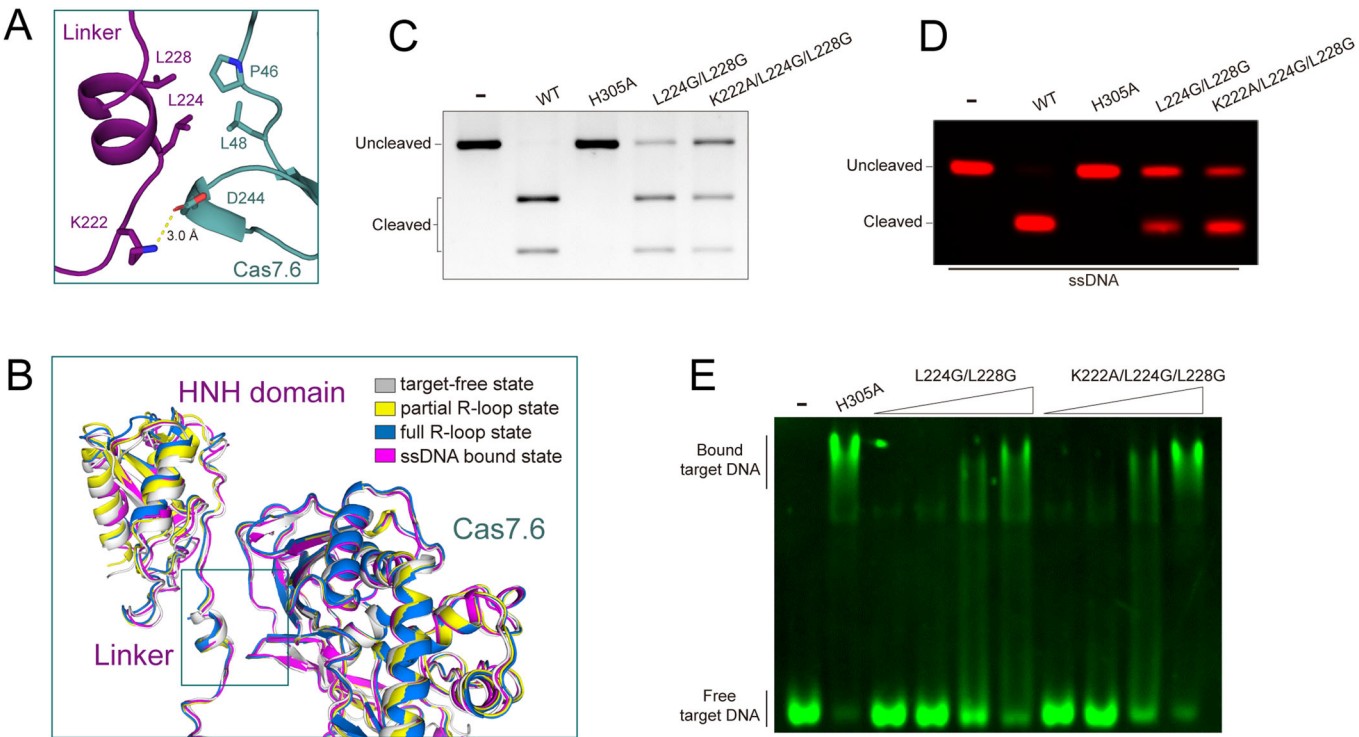

**Figure 3. The Linker part of the HNH domain makes interactions with the Cas7.6 subunit.**

(A) Close-up view of the interaction interface between the Linker of Cas8-HNH protein and Cas7.6 protein. The Linker and Cas7.6 protein at the target engagement state are shown in purple and light teal, respectively. Key interacting residues are shown as sticks. (B) Superposition of Cas8-HNH proteins in target-free (gray) and target-bound states. (C, D) In vitro dsDNA (C) and Cy5-labeled ssDNA (D) cleavage assay of the WT and mutant Cas8-HNH Cascade complex. All experiments were repeated at least three times. (E) The gel electrophoresis mobility shift assay (EMSA) with the Cascade complex harboring Cas8-HNH mutants. The Cy3-labeled target DNA was used as a probe. Source data are available online for this figure.

between the Linker and Cas7.6 subunit are nearly identical across all states, including target-free, partial R-loop, full R-loop, and ssDNA-bound states (Fig. 3B). Additionally, the Linker region within the Cascade complex maintains a rigid position in all the states, indicating that it likely functions as a scaffold for the Cascade complex assembly. The L224G/L228G mutation in this Linker, which could potentially compromise its binding to Cas7.6, impaired cleavage of both dsDNA and ssDNA (Fig. 3C,D), whereas the target binding was not altered (Fig. 3E), highlighting the important role of the linker in positioning the HNH domain for its function. Of note, in the canonical type I-F system, the region equivalent to the Linker is a loop, which undergoes an ordered-to-disordered transition, allowing it to accommodate the rotation of the HB domain upon target DNA binding (Appendix Fig. S4A–C). The flipped HB domain locks the complementary strand of target DNA and acts as a binding platform for the Cas3 helicase-nuclease. This contrasts with the restricted dynamics in the Linker region of the Cas8-HNH system, indicating a unique modulation mechanism for the Cas8-HNH system.

## The target DNA binding induces substantial conformational rearrangement

Compared to target-free states, hinge-bending motions were observed between Cas7 subunits upon target binding. Specifically, in the full R-loop state, Cas7.2 rotates outward around the axis at the Cas7.1-2 interface. Subsequently, Cas7.3 undergoes a similar

outward rotation around the axis at the Cas7.2-3 interface. These consistent rotational movements of adjacent Cas7 subunits result in radial elongation of the Cascade backbone and conformational changes in the crRNA (Appendix Fig. S10A). When aligned with the target-free state at the Cas6 end (crRNA 3′-end), the Cascade complex in the partial R-loop state exhibits a maximum displacement of ~7 Å, while the complexes in the full R-loop and ssDNA-bound states undergo displacements of ~22 Å (Fig. 4A,B; Appendix Fig. S10B–D). These structural observations suggest that the base pairing between the target DNA and crRNA spacer sequence triggers significant conformational changes necessary for the activation of target DNA cleavage by the type I-F Cas8-HNH system, reminiscent of the canonical type I system where Cas3 recruitment only occurs after target DNA binding (Guo et al, 2017; Rollins et al, 2019) (Appendix Fig. S4A–C). We next conducted the target DNA cleavage assays using various mismatched DNA substrates to investigate the DNA mismatch tolerance of the type I-F Cas8-HNH system. Neither mismatches at PAM-proximal positions (for example, 2–4 MM, 8–10 MM, and 14–16 MM) nor mismatches at PAM-distal positions (20–22 MM and 26–28 MM) activated the target DNA cleavage (Fig. 4C). However, it appears that these mismatches do not affect the target DNA binding to the Cascade complex (Fig. 4D), indicating that a correctly formed target DNA-crRNA hybrid, coupled with conformational changes in the Cas7 backbone, is essential for nuclease activity.

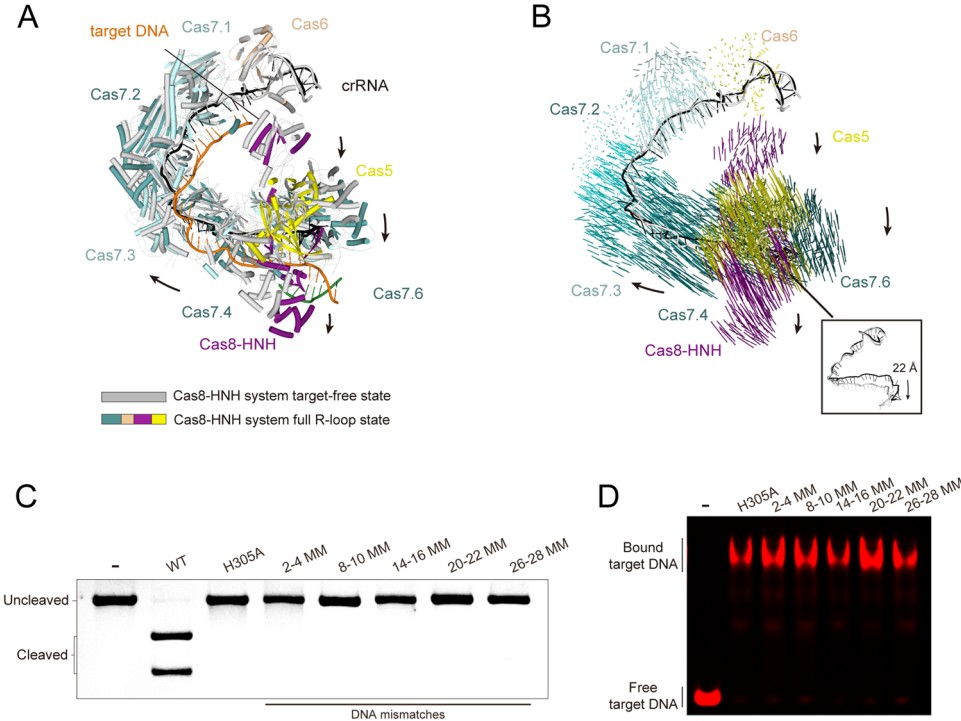

**Figure 4. Full R-loop formation is vital for the activation of the Cas8-HNH system.**

(A) Superposition of the Cas8-HNH complex in target-free and full R-loop states. The arrows indicate the conformational changes in the Cascade backbone and HNH nuclease domain. (B) Vector map of conformational changes in Cas8-HNH Cascade upon full R-loop formation. Domains are colored in the same scheme as in 1C. The insert panel shows the structural comparison of crRNA strands in target-free and full R-loop states. (C) In vitro dsDNA cleavage assay of Cas8-HNH complex with different mismatched dsDNA. The mismatches at the 2nd-4th positions of crRNA are denoted as 2–4 MM (mismatches), and so on. (D) EMSA was performed in the presence of different mismatched dsDNA. The dsDNA was labeled with a Cy5 fluorophore. The experiments were replicated at least three times. Source data are available online for this figure.

## HNH nuclease activation by complete base pairing between the target DNA and crRNA

As mentioned above, the type I-F Cas8-HNH system had no cleavage activity towards non-targeting DNA (Appendix Fig. S1C), suggesting a base-pairing activation mechanism. In the target-free state, the Cas6 subunit contacts with the HNH nuclease domain (Fig. 5A; Appendix Fig. S11A). In particular, Asp252 of the HNH domain forms salt bridges with Arg125 on the Cas6 protein. Ser324 and Tyr301 of the HNH domain are hydrogen bonded with the Gln136 and Lys133 from Cas6 protein, respectively. Apart from the polar interactions, Val299 of the HNH domain makes hydrophobic contact with Phe48 of Cas6 (Appendix Fig. S11A). However, elongation of the Cascade backbone progressively displaces the Cas8-HNH protein relative to the Cas6 subunit upon target binding (Fig. 5A–C). In the partial R-loop state, the HNH nuclease domain undergoes mostly minor movements compared to that in the target-free state, whereas the HNH nuclease moves away from the Cas6 subunit by approximately 11 Å in the full R-loop and ssDNA-bound states, thereby disrupting the HNH-Cas6 interaction interface. As a result, the helix from Cas6 responsible for contacting the HNH domain is not visible in the full R-loop and ssDNA states owing to flexibility (Fig. 5C; Appendix Fig. S11B). These structural changes indicate that HNH displacement from the Cas6 subunit might be crucial for the nuclease activity of this system. Nonetheless, mutations in Cas6 residues that potentially impair

HNH-binding had little impact on target DNA cleavage (Appendix Fig. S11C,D), indicating that Cas6-HNH interaction is not a prerequisite for regulating HNH movement.

The HNH domain cleaves the target DNA at the PAM-distal end (Altae-Tran et al, 2023), which agrees with the location of the HNH domain. However, a mismatch at the PAM-distal region (26th-28th nt, denoted as 26–28 MM), mimicking the partial R-loop state, abolished HNH nuclease activity. This suggests a mechanism for HNH activation that depends on the length of base pairing. In the full R-loop state, the 32-nt spacer sequence is fully complementary to target DNA, and the 32-bp target DNA-crRNA duplex is clearly resolved in the EM density. Our mismatch experiments showed that even mismatches at the PAM-distal 31st-32nd nucleotides (31–32 MM) impeded cleavage of both dsDNA and ssDNA targets but not target binding (Fig. 5D; Appendix Fig. S11E,F), highlighting the importance of faithful target DNA recognition in HNH nuclease activation. Collectively, these results demonstrate that the HNH nuclease activity can be initiated only when the full protospacer sequence on the target DNA is validated.

## The C-end helix of the HNH domain might work as a sensor in response to target DNA binding

Since the target DNA mismatches in the protospacer sequence are not tolerated by the type I-F Cas8-HNH system, we analyzed the

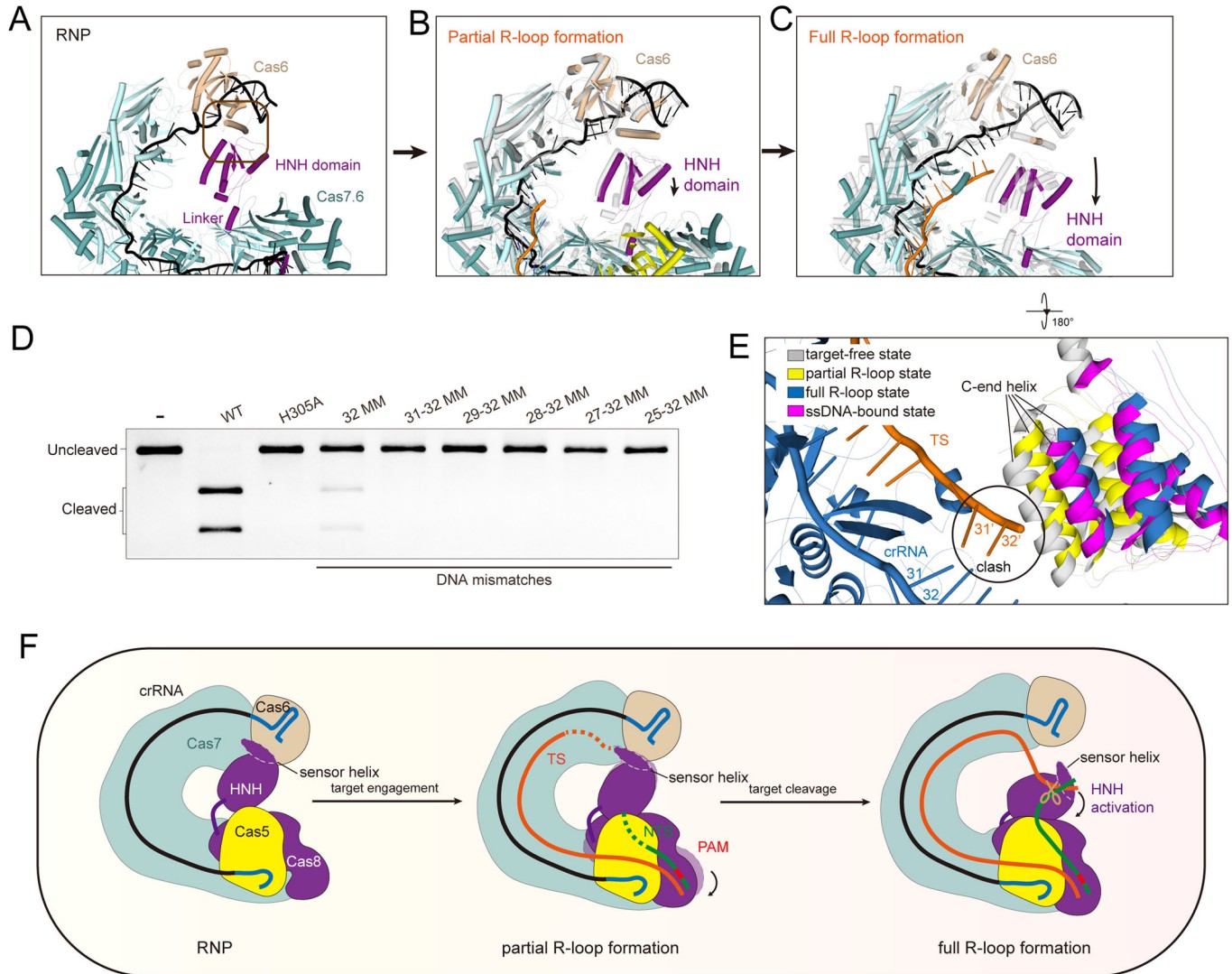

**Figure 5.  The C-terminal helix of the HNH domain modulates the HNH nuclease activity.**

(**A**) The HNH domain (purple) is tethered to the Cas6 subunit in the target-free state. (**B**) Conformational changes of HNH domain upon partial R-loop formation. A slight shift is observed in the HNH domain upon partial R-loop formation. (**C**) Full R-loop formation induces a large rotation of the HNH domain relative to Cas6. The black arrow indicates that the HNH domain shifts away from the Cas6 protein. (**D**) In vitro dsDNA cleavage assay in the presence of mismatched dsDNA. The gel represents three independent experiments. (**E**) Close-up view of the conformational changes of the HNH domain upon target-binding. The possible clashes are indicated by a black circle. Except for the Cascade complex in full R-loop state and the HNH domains, the Cascade complexes in target-free, partial R-loop, and ssDNA-bound states were omitted for clarity. (**F**) Schematic diagram of the activation mechanism of Type I-F Cas8-HNH system. Source data are available online for this figure.

structures to understand the mechanism behind this high fidelity. As aforementioned, the HNH domain swings away from Cas6 and is thought to adopt a catalytically competent conformation in the full R-loop and ssDNA-bound states. Intriguingly, in target-free and partial R-loop states, the C-terminal helix of the HNH domain clashes sterically with the PAM-distal end of the target strand (TS), particularly the nucleotide at 31′-32′ position (Fig. 5E). However, this steric hindrance is likely absent in the 31st-32rd mismatched DNA because the unpaired 31′-32′ nucleotides cannot reach the C-terminal helix, potentially explaining why 31st-32rd mismatches (31–32 MM) prevent the target DNA cleavage (Fig. 5E). Therefore, it is plausible that the C-terminal helix of the

HNH domain might act as a molecular sensor to monitor the integrity of the TS-crRNA hybrid and trigger the conformational rearrangement of HNH domain necessary for cleavage. Mutations within this helix (residues 325–344), such as E329A/T332A/E333A and H335A/K340A/W341A, abolished target DNA cleavage, further supporting its key role in HNH activation (Appendix Fig. S12).

The 33rd-37th nucleotides at the 3′-end repeat sequence, located immediately behind the spacer sequence, are solvent-exposed (Figs. 2B and 5C). It seems that target DNA is possible to base pair with the 5-nt repeat sequence and affects target cleavage. However, the target DNA with a 5′-anti-tag sequence complementary to the

3′-end repeat sequence had little impact on cleavage (Appendix Fig. S13A,B). Additionally, the cleavage site was not altered in the presence of a complementary 5′-anti-tag sequence (Appendix Fig. S13C). These results suggest that the target DNA may not preferentially hybridize with the 3′-end repeat sequence, and the type I-F Cas8-HNH system cleaves target DNA precisely. The observed high target specificity likely originates from the activation mechanism. The spatial restriction of the HNH domain within the Cas8-HNH complex might limit the insertion of target DNA only downstream of the crRNA-DNA hybrid and within a specific length range into the HNH domain's catalytic pocket for cleavage. In the target-bound state, the C-terminal helix is positioned close to the solvent-exposed repeat sequence (Fig. 5C). Therefore, a complementary 5′-anti-tag sequence-mediated extended target DNA-crRNA duplex would not be favored here. Indeed, steric clashes are observed when modeling a base pair between target DNA and 3′-end repeat sequence (Fig. 5E). Together, it is conceivable that the C-terminal helix of the HNH domain has a vital role in orchestrating the activation of the type I-F Cas8-HNH system and its precise target DNA cleavage.

## Discussion

In our study, we structurally and biochemically characterized the newly identified type I-F variant, namely the Cas8-HNH system. The Cascade complex in this Cas8-HNH system exhibits an overall G-shaped architecture, reminiscent of that in the canonical system (Appendix Fig. S4A,B). While the canonical type I Cascade complex recruits Cas3 helicase-nuclease to the Cas8 subunit for target DNA degradation, the Cas8-HNH system features a naturally fused HNH nuclease domain into the Cas8 subunit, replacing the HB domain of the Cas8 subunit in the canonical type I-F members. In the target-free state, the HB domain is centrally located within the Cascade complex by binding to two Cas7 molecules (Appendix Fig. S4A), whereas the HNH domain of the Cas8-HNH system is buttressed by the Cas6 subunit at the head of the Cascade backbone.

Both the typical and Cas8-HNH systems require target DNA binding for activation, although they utilize distinct nuclease effectors and mechanisms. Cas3 recruitment and activation are dependent on the HB domain of Cas8, while the HNH domain is activated directly upon target DNA binding. Target engagement forms the R-loop bubble and induces rearrangements in the Cascade complex for the activation of nuclease activity (Fig. 5F). Full target recognition along the Cascade backbone induces a maximum net axial displacement of 30 Å and allosterically rotates the HB domain of the Cas8 subunit (Appendix Fig. S4C), facilitating Cas3 recruitment. In the Cas8-HNH Cascade complex, full target engagement induces a maximum displacement of only 22 Å in the Cascade backbone, allowing the HNH nuclease to access the target DNA. Hence, a structural state capturing the HNH domain's action on target DNA would be valuable for fully elucidating the activation mechanism.

Unlike the Cas3 helicase-nuclease in canonical type I systems that induces NTS nicking and processive target DNA degradation in an ATP-dependent manner (Xiao et al, 2018), the HNH nuclease domain

generates a double-strand break in the target dsDNA molecule and provides a precise crRNA-guided dsDNA cleavage independently of ATP. Therefore, it is plausible that the HNH domain functions as a single turnover enzyme. On the other hand, the Cas8-HNH system can also target and cut ssDNA in addition to dsDNA substrates, expanding its potential applications to ssDNA-targeting. To investigate how the Cas8-HNH system generates a double-strand break, particularly the mechanism by which the NTS passes through the Cascade complex to reach the HNH catalytic pocket, we observed a positively charged groove around the Cas8-HNH protein, implying a potential NTS binding channel (Appendix Fig. S14A,B). Mutation of the key residues in this channel, such as K207A/W208A, impaired target DNA binding, and cleavage (Appendix Fig. S14C–E), demonstrating the critical role of the potential NTS binding groove in HNH-mediated target cleavage.

The type II CRISPR-Cas9 systems, which are the most widely used in gene editing (Du et al, 2023; Wang et al, 2022), tolerate DNA mismatches at the PAM-distal end, leading to the off-target cleavage effects (Pacesa et al, 2022). According to the mismatched DNA cleavage assay, the HNH nuclease activity is not activated unless the entire protospacer sequence of the target DNA is validated. Even a single nucleotide mismatch in the PAM-distal sequence fails to activate the target DNA cleavage by the type I-F Cas8-HNH system, suggesting the potential for an accurate DNA editing tool. Furthermore, the Cas8-HNH system specifically identifies a CN PAM sequence, which is more flexible than the CC PAM in the canonical Type I-F system and the NGG PAM for *Streptococcus pyogenes* Cas9 (SpCas9), indicating a broader targeting range. Together, our structural and biochemical findings expand the potential applications of type I CRISPR systems in gene editing, laying a foundation for the development of efficient and highly precise DNA editing tools.

## Methods

### Reagents and tools table

| Reagent/Resource | Reference or source | Identifier or catalog number |
|---|---|---|
| **Experimental Models** | | |
| Escherichia coli strain DH5α | Tsingke Biotechnology | TSC-C14 |
| Escherichia coli BL21 (DE3) | Tsingke Biotechnology | TSC-E01 |
| **Recombinant DNA** | | |
| pCDFDuet-1_Cas8/Cas5 vector | This study | |
| pRSFDuet-1_Cas7/Cas6 vector | This study | |
| pACYCT2_type I-F Cas8-HNH crRNA array | This study | |
| **Oligonucleotides and other sequence-based reagents** | | |
| Cas8-HNH system crRNA for cryo-EM study | This study | UUUAGAAGGAUUG CCGUCAGGAAAUU AGGUGCGCUUAGC AGUGUACCGCCGGA UAGGCGG |
| Cas8-HNH system NTS for structure determination | This study | AGCTGGTGC GGATTGCCG TCAGGAAATT AGGTGCGCTT AGCACACATG TCAAGCTC |

| Reagent/Resource | Reference or source | Identifier or catalog number |
|---|---|---|
| Cas8-HNH system TS for structure determination | This study | GAGCTTGACA TGTGTGCTAA GCGCACCTAA TTTCCTGACGG CAATCCGCACCAGCT |
| Cas8-HNH system ssDNA for structure determination | This study | GAGCTTGACAT GTGTGCTAAGCG CACCTAATTTCCT GACGGCAATCCGC ACCAGCT |
| **Chemicals, enzymes, and other reagents** | | |
| HiTrap Heparin HP column | Cytiva | 17040701 |
| Superose™ 6 Increase 10/300 GL | Cytiva | 29091596 |
| **Software** | | |
| Cryosparc | Punjani et al (2017) | |
| AlphaFold2 | Jumper et al (2021) | |
| Phenix | Williams et al (2018) | |
| Coot | Emsley and Cowtan (2004) | |
| PyMOL | https://pymol.org/ | |
| UCSF ChimeraX | Pettersen et al (2021) | |
| **Other** | | |
| ÄKTA pure | Cytiva | N/A |
| Visible Fluorescent Imager | Azure biosystems | N/A |

## Protein expression and purification

The genes encoding full-length Cas5, Cas6, Cas7, and Cas8-HNH proteins of the type I-F Cas8-HNH system from *Selenomonas* sp. isolate RGIG9219 were codon optimized and synthesized by GENEWIZ. The gene encoding Cas5 and N-terminal His$_6$-tagged Cas8-HNH were amplified by PCR and cloned into a pCDFDuet vector. Moreover, Cas7 and Cas6 genes were cloned into the pRSFDuet-1 vector, and the gene encoding a 68-nt crRNA was cloned into a pACYCT2 vector. All plasmids were co-transformed into *Escherichia coli* BL21 (DE3) cells. The monoclonal colony was picked and cultured in Luria broth (LB) medium at 37 °C. When the optical density (OD$_{600}$) value reached 0.6, the protein expression was induced by 0.2 mM isopropyl-β-D-thiogalactoside (IPTG), and cells were growing for 12–16 h at 20 °C. The harvested cells were resuspended in the binding buffer (25 mM Tris-HCl pH 7.5, 500 mM NaCl, 3 mM β-mercaptoethanol, 5 mM imidazole, and 1% glycerol) and were lysed by sonication on ice. Next, the lysate was centrifuged at 14,000 rpm for 40 min, and Ni-NTA resin (Qiagen) was added to the clarified lysate and incubated on ice for 1 h. For the purification of the Cas8-HNH Cascade-crRNA binary complex, the resin was washed extensively with binding buffer three times, and eluted with the binding buffer supplementary with 300 mM imidazole. The eluate was loaded onto a HiTrap Heparin HP column (Cytiva). Peak fractions containing the target protein complex were concentrated and further purified by a Superdex 6 column (Cytiva) equilibrated with the gel-filtration buffer (25 mM Tris-HCl pH 7.5, 200 mM NaCl, and 2 mM DTT). Peak fractions

were collected, and the SDS-PAGE gel was utilized to analyze the purified protein complex. The mutants of the Cas8-HNH complex were purified in the same way as the wild-type proteins.

## Cryo-EM sample preparation and data processing

About 4 μl of purified Cas8-HNH Cascade-crRNA complex and Cascade-crRNA-target DNA complex were applied to the glow-discharged gold grids covered with cryoMatrix amorphous Ni-Ti alloy film (R1.2/1.3) or GIG gold grids (R1.2/1.3). The grids were then blotted for 2 s at 4 °C in 100% humidity, followed by vitrification by plunging into liquid ethane which was cooled by liquid nitrogen in a Vitrobot (FEI). The datasets of Cas8-HNH complex in target-free, dsDNA-bound, and ssDNA-bound states were collected on a 300 kV Titan Krios microscope (FEI) with a pixel size of 0.856, 0.75, or 0.808 Å, respectively. Data collection was collected by SerialEM (Mastronarde, 2005) with the defocus range of −1.2 to −2.0 μm. Each micrograph was dose-fractioned to 32 frames. The total dose was 60.0 e$^-$/Å$^2$ per micrograph.

The cryo-EM data analysis was commenced with the beam-induced motion by MotionCor2 (Zheng et al, 2017). The CTF estimation was performed using Gctf (Zhang, 2016), and the particle picking was executed in CryoSPARC (Punjani et al, 2017). For the dataset of the Cas8-HNH complex at a target-free state, 750,550 particles were picked by blob-picker in CryoSPARC (Punjani et al, 2017). After several rounds of 2D classification, about 150k particles were kept for the following ab initio reconstruction and topaz particle picking. Approximately 319k particles were picked by topaz and were directly used for the subsequent heterogeneous refinement. Two classes with clear HNH domain features which contained 220,116 particles, were combined for one round of non-uniform refinement. The final resolution of the Cas8-HNH complex at target-free state was at 3.6 Å.

For the dataset of Cas8-HNH Cascade complex in target-bound states, 8796 micrographs were collected, and 1,645,462 particles were picked by blob-picker in CryoSPARC (Punjani et al, 2017). Several rounds of heterogeneous refinement were performed to screen these good particles. About 714k good particles featured with a clear density of HNH domain were kept. Another round of 2D classification was further performed to sort out the good particles. One round of heterogeneous refinement was then performed and two classes with different HNH conformations, named class 1 and class 2, were captured. Non-uniform refinement was performed to push the resolution, and the final resolution for class 1 and class 2 were at 2.51 and 2.57 Å, respectively.

For the dataset of Cas8-HNH Cascade complex in ssDNA-bound state, imaging processing was carried out almost as previously described. 439,319 particles were picked by blob-picker in CryoSPARC. 2D classification, ab initio reconstruction and non-uniform refinement were then performed to generate a map at 2.90 Å.

## Structure determination

The initial atomic models of the Cas5, Cas6, Cas7, and Cas8-HNH subunits predicted by AlphaFold2 were docked manually into the cryo-EM density map using ChimeraX (Pettersen et al, 2021). For modeling of the RNP complex, docked models of the Cas8-HNH

complex were utilized for crRNA density segmentation and the nucleotides were built into crRNA density manually in Coot (Emsley and Cowtan, 2004). For modeling of target-bound Cas8-HNH complex at the partial and full R-loop formation states, the protein subunits and crRNA from the refined structure at the target-free state were separately docked into the EM density in ChimeraX (Pettersen et al, 2021). Target DNA was built manually according to the EM density map. Further iterative model refinements were performed by *phenix.real_space_refine* (Williams et al, 2018) and Coot (Emsley and Cowtan, 2004). The quality of the final models was validated by MolProbity in Phenix (Williams et al, 2018). Data collection and model refinement statistics are concluded in Appendix Table S1.

### In vitro DNA cleavage assay

The 60-bp target DNA sequence was cloned into a pET-derived vector. For the target DNA cleavage assay, the resultant plasmid was linearized by FspI (NEB). The Cas8-HNH complex and dsDNA substrate were prediluted by the cleavage buffer containing 50 mM Tris-HCl, pH 7.5, 100 mM NaCl, 5 mM $MgCl_2$, and 1 mM DTT. To test the cleavage activity of the Type I-F Cas8-HNH system, the dsDNA target was incubated with WT or mutant Cas8-HNH complex at molar ratios ranging from 1:10 to 1:50. Reaction was conducted at 37 °C for 1 h. About 40 mM EDTA and 1 mg/ml proteinase K were added to terminate the reaction, and the cleavage product was visualized by 0.5% TBE agarose gel with StarStain Red Nucleic Acid Dye (GenStar). The same procedure was conducted as above for in vitro dsDNA cleavage assay of Cas8-HNH protein and HNH domain.

### In vitro ssDNA cleavage assay

The 3′-Cy5 labeled and 5′-Cy3-labeled ssDNA substrates were synthesized by GENWIZ. The Cas8-HNH complex and ssDNA were prediluted by the cleavage buffer (50 mM Tris-HCl, pH 7.5, 100 mM NaCl, 5 mM $MgCl_2$, and 1 mM DTT). The 3′-Cy5 labeled targeting ssDNA was incubated with Cas8-HNH complex at molar ratios ranging from 1:0.1 to 1:10, and the reaction was conducted at 37 °C for 30 min. The reaction products were separated by 15% urea gel.

For the ssDNA cleavage assay of 5′-Cy3-labeled non-targeting ssDNA, the ssDNA molecule was incubated with WT or mutant Cas8-HNH complex at a molar ratio of 1:30. Reaction was conducted at 37 °C for 1 h and stopped by 40 mM EDTA and 1 mg/ml protease K. Reaction products were visualized as above.

### Electrophoretic mobility shift assay (EMSA)

The binding of target DNA to the Cas8-HNH complex was analyzed with DNA EMSA. The TS and 5′-Cy3-labeled NTS were annealed by heating up to 95 °C for 3 min, and gradually cooling to 25 °C. The purified Cas8-HNH complex was incubated with annealed dsDNA at 37 °C for 30 min in the buffer containing 50 mM Tris-HCl, pH 7.5, 10 mM NaCl, and 1 mM DTT with an increasing concentration of WT or mutants Cas8-HNH complex. All samples were loaded onto a 5% native PAGE gel and running for 30 min at 120 V at 4 °C. Reaction products were visualized using a Tanon 5200 imaging system.

## Data availability

The atomic coordinates and EM maps have been deposited in the Protein Data Bank (https://www.rcsb.org/) under accession codes 8ZDY (RNP complex), 8ZNR (ssDNA-bound state), 8Z0K (full R-loop state), and 8Z0L (partial R-loop state), and in the Electron Microscopy Data Bank (https://www.ebi.ac.uk/emdb/) under corresponding accession codes EMD-60017, EMD-60279, EMD-39706, and EMD-39707.

The source data of this paper are collected in the following database record: biostudies:S-SCDT-10_1038-S44318-024-00229-8.

## Peer review information

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

## Acknowledgements

This work was supported by the National Natural Science Foundation of China (32322040 to HZ and 32300036 to HY).

## Author contributions

**Xuzichao Li**: Formal analysis; Investigation; Writing—original draft; Writing—review and editing. **Yanan Liu**: Formal analysis; Investigation; Methodology. **Jie Han**: Validation; Investigation; Methodology. **Lingling Zhang**: Formal analysis; Investigation; Methodology. **Zhikun Liu**: Software; Formal analysis; Visualization; Methodology. **Lin Wang**: Software; Formal analysis; Investigation; Methodology. **Shuqin Zhang**: Software; Validation; Investigation; Visualization. **Qian Zhang**: Investigation; Methodology. **Pengyu Fu**: Investigation. **Hang Yin**: Formal analysis; Funding acquisition. **Hongtao Zhu**: Conceptualization; Resources; Supervision; Methodology; Project administration; Writing—review and editing. **Heng Zhang**: Conceptualization; Resources; Supervision; Funding acquisition; Validation; Visualization; Writing—original draft; Project administration; Writing—review and editing.

Source data underlying figure panels in this paper may have individual authorship assigned. Where available, figure panel/source data authorship is listed in the following database record: biostudies:S-SCDT-10_1038-S44318-024-00229-8.

## Disclosure and competing interests statement

The authors declare no competing interests.

