## [Peer Review File · The EMBO Journal]

Structural basis for the type I-F Cas8-HNH system

Heng Zhang, xuzichao li, yanan liu, Jie Han, lingling zhang, zhikun liu, lin wang, shuqin zhang, qian zhang, pengyu fu, Hang Yin, and hongtao zhu

Corresponding author(s): Heng Zhang (zhangheng134@gmail.com) , hongtao zhu (hongtao.zhu@iphy.ac.cn)

Review Timeline:

Submission Date:	5th Aug 24
Editorial Decision:	9th Aug 24
Revision Received:	21st Aug 24
Accepted:	27th Aug 24

Editor: Hartmut Vodermaier

Transaction Report:

This manuscript was transferred to The EMBO JOURNAL following peer review at another journal.

Dr. Heng Zhang
TianJin Medical University
No22 Qixiangtai Rd
TianJin 300070
China

9th Aug 2024

Re: EMBOJ-2024-118682-T
Structural basis for the type I-F Cas8-HNH system

Dear Dr. Zhang,

Thank you for transferring your previously reviewed manuscript, together with referee reports and revision plans, to The EMBO Journal. As already communicated in our transfer offer, we would be interested in publishing your study without further experimental revisions, pending making the initially missing EM maps available for checking by the original referee 2. I am happy to say that the referee has now assessed the maps and found them generally satisfactory (see comments below). I am therefore inviting you to revise the manuscript based on your tentative point-by-point response and according to EMBO Journal editorial policies, as follows:

Scientific points:

- Please incorporate the clarifications and revisions in response to referee 1's minor suggestions, as already outlined in your revision plan.
- For referee 2, please do include the proposed more comprehensive description and discussion of differences between divergent CRISPR systems; while additional reconstitution, crosslinking, or structural experiments (including attempts to capture target DNA-inserted structures) shall not be required.
- For referee 2's specific points 2-5 and 7-8, please again incorporate the responses already provided in the revision plan. Should you in the meantime already have obtained conclusive data to answer additional queries, e.g. point 6, please feel free to add them to the revised version as well, but it would not be essential.

Editorial points:

- Please upload all main Figures as individual files with sufficient resolution/quality for production, separate from the main text file.
- Please carefully complete the attached Author Checklist and upload it at the time of resubmission together with the revised manuscript files.
- If at all possible, please supply institutional email addresses (rather than freemail addresses) for all corresponding authors.
- "Supplementary" Figures & Tables: Please rename them to "Appendix Figure S1/2/3..." and "Appendix Table S1" both when referencing them from the main text, and in the legends/labels. The file containing them should be in PDF format and called "Appendix"; it should be pre-faced by a Table of Contents page saying "Appendix for [article title], listing the authors, and listing the included Appendix Figures and Tables with their respective page numbers.
- On the abstract page of the manuscript, please include 4-5 general keyword terms to enhance searchability.
- Please adjust the format of the reference list and of the in-text citations according to EMBO Journal format (alphabetical order, author name et al + year...)
- All Materials and Methods need to be described in the main text using our 'Structured Methods' format. The Methods section should include a Reagents and Tools Table (downloadable at <https://www.embopress.org/page/journal/14693178/authorguide#structuredmethods>) listing key reagents, experimental models, software and relevant equipment, and including their sources and relevant identifiers; followed by a Methods and Protocols section describing the methods (ideally using a step-by-step protocol format to facilitate adoption of the methodologies across labs)
- In the Data Availability section, please make sure to include direct hyperlinks to each database (PDB, EMDB) in which data

generated in the study have been deposited (and make sure to initiate timely public release upon acceptance).

- Please rename the Conflict of Interest section into "Disclosure and Competing Interests Statement", in accordance with our updated Guide to Authors (<https://www.embopress.org/competing-interests>)
- As we are switching from a free-text author contribution statement towards a more formal statement based on Contributor Role Taxonomy (CRediT) terms, please remove the present Author Contribution section and instead specify each author's contribution(s) directly in the Author Information page of our submission system during upload of the final manuscript. See <https://casrai.org/credit/> for more information.
- Please provide suggestions for a short 'blurb' text prefacing and summing up the conceptual aspect of the study in two sentences (max. 250 characters), followed by 3-5 one-sentence 'bullet points' with brief factual statements of key results of the paper; they will form the basis of an editor-written 'Synopsis' accompanying the online version of the article. Please also upload a synopsis image, which can be used as a "visual title" for the synopsis section of your paper. The image (maybe based on Figure 5F) should be in PNG or JPG format, and please make sure that it remains in the modest dimensions of (EXACTLY) 550 PIXELS WIDE and 300-600 PIXELS HIGH.
- Finally, you shall also receive a separate message from our Source Data curation team, with instructions on how to prepare and upload relevant image and numerical raw data.

Once we have received and checked the final revised version and files, we should be ready to swiftly proceed with formal acceptance and production of the paper. Thank you again for the opportunity to consider this work for The EMBO Journal!

With kind regards,

Hartmut Vodermaier

- 1) Every manuscript requires a Data Availability section (even if only stating that no deposited datasets are included). Primary datasets or computer code produced in the current study have to be deposited in appropriate public repositories prior to resubmission, and reviewer access details provided in case that public access is not yet allowed. Further information: embopress.org/page/journal/14602075/authorguide#dataavailability
- 2) Each figure legend must specify
 - size of the scale bars that are mandatory for all micrograph panels
 - the statistical test used to generate error bars and P-values
 - the type error bars (e.g., S.E.M., S.D.)
 - the number (n) and nature (biological or technical replicate) of independent experiments underlying each data point
 - Figures may not include error bars for experiments with $n < 3$; scatter plots showing individual data points should be used instead.
- 3) Revised manuscript text (including main tables, and figure legends for main and EV figures) has to be submitted as editable text file (e.g., .docx format). We encourage highlighting of changes (e.g., via text color) for the referees' reference.
- 4) Each main and each Expanded View (EV) figure should be uploaded as individual production-quality files (preferably in .eps, .tif, .jpg formats). For suggestions on figure preparation/layout, please refer to our Figure Preparation Guidelines: <http://bit.ly/EMBOPressFigurePreparationGuideline>
- 5) Point-by-point response letters should include the original referee comments in full together with your detailed responses to them (and to specific editor requests if applicable), and also be uploaded as editable (e.g., .docx) text files.
- 6) Please complete our Author Checklist, and make sure that information entered into the checklist is also reflected in the manuscript; the checklist will be available to readers as part of the Review Process File. A download link is found at the top of our Guide to Authors: embopress.org/page/journal/14602075/authorguide

9) To facilitate reproducibility and cross-laboratory adoption of methodologies, please structure the Materials & Methods section as outlined in our guide to authors, including a completed Reagents and Tools Table that can be downloaded from our author guidelines as well (<https://www.embopress.org/page/journal/14602075/authorguide#structuredmethods>).

10) Digital image enhancement is acceptable practice, as long as it accurately represents the original data and conforms to community standards. If a figure has been subjected to significant electronic manipulation, this must be clearly noted in the figure legend and/or the 'Materials and Methods' section. The editors reserve the right to request original versions of figures and the original images that were used to assemble the figure. Finally, we generally encourage uploading of numerical as well as gel/blot image source data; for details see: embopress.org/page/journal/14602075/authorguide#sourcedata

At EMBO Press, we ask authors to provide source data for the main manuscript figures. Our source data coordinator will contact you to discuss which figure panels we would need source data for and will also provide you with helpful tips on how to upload and organize the files.

In the interest of ensuring the conceptual advance provided by the work, we recommend submitting a revision within 3 months (7th Nov 2024). Please discuss the revision progress ahead of this time with the editor if you require more time to complete the revisions. Use the link below to submit your revision:

Link Not Available

Referee #2:

This referee has carefully checked the EM maps, especially the map of 80ZL (the full R-loop structure). The qualities of these EM maps are high. Although it would not influence the major conclusions of the manuscript, the local conformations of several side chains in 80ZL can be further refined/reviced.

Referee #2:

This referee has carefully checked the EM maps, especially the map of 80ZL (the full R-loop structure). The qualities of these EM maps are high. Although it would not influence the major conclusions of the manuscript, the local conformations of several side chains in 80ZL can be further refined/reviced.

RE: We are so grateful for the reviewer's positive comments, the local conformations of side chains in 80ZL have been further revised.

Dr. Heng Zhang
TianJin Medical University
No22 Qixiangtai Rd
TianJin 300070
China

27th Aug 2024

Re: EMBOJ-2024-118682R
Structural basis for the type I-F Cas8-HNH system

Dear Dr. Zhang,

Thank you for submitting your final revised manuscript for our consideration. I am pleased to inform you that we have now accepted it for publication in The EMBO Journal.

Yours sincerely,

Hartmut Vodermaier
